# Neural Power Units

**Niklas Heim, Tomáš Pevný, Václav Šmídl**
Artificial Intelligence Center
Czech Technical University
Prague, CZ 120 00
{niklas.heim, tomas.pevny, vasek.smidl}@aic.fel.cvut.cz

## Abstract

Conventional Neural Networks can approximate simple arithmetic operations, but fail to generalize beyond the range of numbers that were seen during training. *Neural Arithmetic Units* aim to overcome this difficulty, but current arithmetic units are either limited to operate on positive numbers or can only represent a subset of arithmetic operations. We introduce the *Neural Power Unit* (NPU).[1] that operates on the *full domain of real numbers* $\mathbb{R}$ and is capable of *learning arbitrary power functions* in a single layer. The NPU thus fixes the shortcomings of existing arithmetic units and extends their expressivity. We achieve this by using complex arithmetic without requiring a conversion of the network to complex numbers $\mathbb{C}$. A simplification of the unit to the *RealNPU* yields a highly transparent model. We show that the NPUs outperform their competitors in terms of accuracy and sparsity on artificial arithmetic datasets, and that the RealNPU can discover the governing equations of a dynamical system only from data.

## 1   Introduction

Numbers and simple algebra are essential not only to human intelligence but also to the survival of many other species [Dehaene, 2011, Gallistel, 2018]. A successful, intelligent agent should, therefore, be able to perform simple arithmetic. State of the art neural networks are capable of learning arithmetic, but they fail to extrapolate beyond the ranges seen during training [Suzgun et al., 2018, Lake and Baroni, 2018]. The inability to generalize to unseen inputs is a fundamental problem that hints at a lack of *understanding* of the given task. The model merely memorizes the seen inputs and fails to abstract the true learning task. The failure of numerical extrapolation on simple arithmetic tasks has been shown by Trask et al. [2018], who also introduced a new class of *Neural Arithmetic Units* with good extrapolation performance on some arithmetic tasks.

Including Neural Arithmetic Units in standard neural networks promises to significantly increase their extrapolation capabilities due to their inductive bias towards numerical computation. This is especially important for tasks in which the data generating process contains mathematical relationships. They also promise to reduce the number of parameters needed for a given task, which can improve the explainability of the model. We demonstrate this in a *Neural Ordinary Differential Equation* (NODE, Chen et al. [2019]), where a handful of neural arithmetic units can outperform a much bigger network built from dense layers (Sec. 4.1). Moreover, our new unit can be used to directly read out the correct generating ODE from the fitted model. This is in line with recent efforts to build *transparent* models instead of attempting to explain black-box models [Rudin, 2019], like conventional neural networks. We refer to the terminology by Lipton [2017] which defines the potential of understanding the parameters of a given model as *transparency by decomposability*.

The currently available arithmetic units all have different strengths and weaknesses, but none of them solve simple arithmetic completely. The *Neural Arithmetic Logic Unit* (NALU) by Trask et al. [2018], chronologically, was the first arithmetic unit. It can solve addition ($+$, including subtraction), multiplication ($\times$), and division ($\div$), but is limited to positive inputs. The convergence of the NALU is quite fragile due to an internal gating mechanism between addition and multiplication paths as well as the use of a logarithm which is problematic for small inputs. Recently, Schlör et al. [2020] introduced the *improved NALU* (iNALU, to fix the NALU's shortcomings. It significantly increases its complexity, and we observe only a slight improvement in performance. Madsen and Johansen [2020] solve ($+, \times$) with two new units: the *Neural Addition Unit* (NAU), and the *Neural Multiplication Unit* (NMU). Instead of gating between addition and multiplication paths, they are separate units that can be stacked. They can work with the full range of real numbers, converge much more reliably, but cannot represent division.

**Our Contributions**

**Neural Power Unit.** We introduce a new arithmetic layer (NPU, Sec. 3) which is capable of learning products of power functions ($\prod x_i^{w_i}$) of arbitrary real inputs $x_i$ and power $w_i$, thus including multiplication ($x_1 \times x_2 = x_1^1 x_2^1$) as well as division ($x_1 \div x_2 = x_1^1 x_2^{-1}$). This is achieved by using formulas from complex arithmetic (Sec. 3.1). Stacks of NAUs and NPUs can thus learn the full spectrum of simple arithmetic operations.

**Convergence improvement.** We address the known convergence issues of neural arithmetic units by introducing a *relevance gate* that smooths out the loss surface of the NPU (Sec. 3.2). With the relevance gate, which helps to learn to ignore variables, the NPU reaches extrapolation errors and sparsities that are on par with the NMU on ($\times$) and outperforms NALU on ($\div, \sqrt{\cdot}$).

**Transparency.** We show how a power unit can be used as a highly transparent[2] model for equation discovery of dynamical systems. Specifically, we demonstrate its ability to identify a model that can be interpreted as a SIR model with fractional powers (Sec. 4.1) that was used to fit the COVID-19 outbreak in various countries [Taghvaei et al., 2020].

## 2 Related Work

Several different approaches to automatically solve arithmetic tasks have been studied in recent years. Approaches include Neural GPUs [Kaiser and Sutskever, 2016], Grid LSTMs [Kalchbrenner et al., 2016], Neural Turing Machines [Graves et al., 2014], and Neural Random Access Machines [Kurach et al., 2016]. They solve tasks like binary addition and multiplication, or single-digit arithmetic. The Neural Status Register [Faber and Wattenhofer, 2020] focuses on control flow. The Neural Arithmetic Expression Calculator [Chen et al., 2018], a hierarchical reinforcement learner, is the only method that solves the division problem, but it operates on character sequences of arithmetic expressions. Related is symbolic integration with transformers [Lample and Charton, 2019]. Unfortunately, most of the named models have severe problems with extrapolation [Madsen and Johansen, 2019, Saxton et al., 2019]. A solution to the extrapolation problem could be Neural Arithmetic Units. They are designed with an inductive bias towards systematic, arithmetic computation. However, currently, they are limited in their capability of expressing the full range of simple arithmetic operations ($+, \times, \div$). In the following two sections, we briefly describe the currently available arithmetic layers, including their advantages and drawbacks.

### 2.1 Neural Arithmetic Logic Units

Trask et al. [2018] have demonstrated the severity of the extrapolation problem of dense networks for even the simplest arithmetic operations, such as summing or multiplying two numbers. To increase the power of abstraction for arithmetic tasks, they propose the *Neural Arithmetic Logic Unit* (NALU), which is capable of learning ($+, \times, \div$). However, the NALU cannot handle negative inputs correctly due to the logarithm in Eq. 2:

**Definition** (**NALU**). *The NALU consists of a (+) and a (×) path with shared weights* $\boldsymbol{W}$ *and* $\boldsymbol{M}$.

$$\text{Addition: } \boldsymbol{a} = \hat{\boldsymbol{W}}\boldsymbol{x} \qquad\qquad \hat{\boldsymbol{W}} = \tanh(\boldsymbol{W}) \odot \sigma(\boldsymbol{M}) \qquad (1)$$

$$\text{Multiplication: } \boldsymbol{m} = \exp(\hat{\boldsymbol{W}}\log(|\boldsymbol{x}| + \epsilon)) \qquad\qquad\qquad\qquad\quad (2)$$

$$\text{Output: } \boldsymbol{y} = \boldsymbol{a} \odot \boldsymbol{g} + \boldsymbol{m} \odot (1 - \boldsymbol{g}) \qquad\qquad \boldsymbol{g} = \sigma(\boldsymbol{G}\boldsymbol{x}) \qquad (3)$$

*with inputs* $\boldsymbol{x}$ *and learnt parameters* $\boldsymbol{W}$, $\boldsymbol{M}$, *and* $\boldsymbol{G}$.

Additionally, the logarithm destabilizes training to the extent that the chance of success can drop below 20% for $(+, \times)$, it becomes practically impossible to learn $(\div)$ and difficult to learn from small inputs in general [Madsen and Johansen, 2019]. Schlör et al. [2020] provide a detailed description of the shortcomings of the NALU, and they suggest an *improved NALU* (iNALU). The iNALU addresses the NALU's problems through several mechanisms. It has independent addition and multiplication weights for Eq. 1 and Eq. 2, clips weights and gradients to improve training stability, regularizes the weights to push them away from zero, and, most importantly, introduces a mechanism to recover the sign that is lost due to the absolute value in the logarithm. Additionally, the authors propose to reinitialize the network if its loss is not improving during training. We include the iNALU in one of our experiments and find that it only slightly improves the NALU's performance (Sec. 4.2) at the cost of a significantly more complicated unit. Our NPU avoids all these mechanisms by internally using complex arithmetic.

## 2.2 Neural Multiplication Unit & Neural Addition Unit

Instead of trying to fix the NALU's convergence issues, Madsen and Johansen [2020] propose a new unit for $(\times)$ only. The *Neural Multiplication Unit* (NMU) uses explicit multiplications and learns to gate between identity and $(\times)$ of inputs. The NMU is defined by Eq. 4 and is typically used in conjunction with the so-called *Neural Addition Unit* (NAU) in Eq. 5.

**Definition** (**NMU & NAU**). *NMU and NAU are two units that can be stacked to model* $(+, \times)$.

$$\text{NMU: } y_j = \prod_i \hat{M}_{ij}x_i + 1 - \hat{M}_{ij} \qquad\qquad \hat{M}_{ij} = \min(\max(M_{ij}, 0), 1) \qquad (4)$$

$$\text{NAU: } \boldsymbol{y} = \hat{\boldsymbol{A}}\boldsymbol{x} \qquad\qquad \hat{A}_{ij} = \min(\max(A_{ij}, -1), 1) \qquad (5)$$

*with inputs* $\boldsymbol{x}$, *and learnt parameters* $\boldsymbol{M}$ *and* $\boldsymbol{A}$.

Both NMU and NAU are regularized with $\mathcal{R} = \sum_{ij} \min(|W_{ij}|, |1 - W_{ij}|)$, and their weights are clipped, which biases them towards learning an operation or pruning it completely. The combination of NAU and NMU can thus learn $(+, \times)$ for both positive and negative inputs. Training NAU and NMU is stable and succeeds much more frequently than with the NALU, but they cannot represent $(\div)$, which we address with our NPU.

# 3 Neural Power Units

To fix the deficiencies of current arithmetic units, we propose a new arithmetic unit (inspired by NALU) that can learn arbitrary products of power functions ($\prod x_i^{w_i}$) (including $\times, \div$) for positive and negative numbers, and still train well. Combined with the NAU, we solve the full range of arithmetic operations. This is possible through a simple modification of the $(\times)$-path of the NALU (Eq. 6). We suggest to replace the logarithm of the absolute value by the complex logarithm and to allow $\boldsymbol{W}$ to be complex as well. Since the complex logarithm is defined for negative inputs, the NPU does not have a problem with negative numbers. A complex $\boldsymbol{W}$ improves convergence at the expense of transparency (see Sec. 4.1). The improvement during training might be explained by the additional imaginary parameters that make it possible to avoid regions with an uninformative gradient signal.

## 3.1 Naive Neural Power Unit – NaiveNPU

With the modifications introduced above we can extend the multiplication path of the NALU from

$$\boldsymbol{m} = \exp(\boldsymbol{W}\log_{\text{real}}(|\boldsymbol{x}| + \epsilon)) \qquad\qquad\qquad\qquad (6)$$

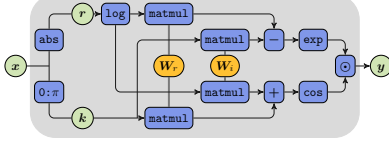

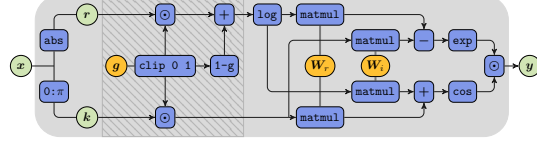

Figure 1: NaiveNPU diagram, with input $\boldsymbol{x}$ and output $\boldsymbol{y}$. Vectors in green, trainables in orange, functions in blue.

Figure 2: NPU diagram. The NPU has a relevance gate $\boldsymbol{g}$ (hatched background) in front of the input to the unit to prevent zero gradients.

to use the complex logarithm ($\log := \log_{\text{complex}}$) and a complex weight $\boldsymbol{W}$ to

$$\boldsymbol{y} = \exp(\boldsymbol{W} \log \boldsymbol{x}) = \exp\left((\boldsymbol{W}_r + i\boldsymbol{W}_i) \log \boldsymbol{x}\right), \tag{7}$$

where the input $\boldsymbol{x}$ is still a vector of real numbers. With the polar form for a complex number $z = re^{i\theta}$ the complex log applied to a real number $x = re^{ik\pi}$ is

$$\log x = \log r + ik\pi, \tag{8}$$

where $k = 0$ if $x \geq 0$ and $k = 1$ if $x < 0$. Using the complex log in Eq. 7 lifts the positivity constraint on $\boldsymbol{x}$, resulting in a layer that can process both positive and negative numbers correctly. A complex weight matrix $\boldsymbol{W}$ somewhere in a larger network would result in complex gradients in other layers. This would effectively result in doubling the number of parameters of the whole network. As we are only interested in real outputs, we can avoid this doubling by considering only the real part of the output $\boldsymbol{y}$:

$$\text{Re}(\boldsymbol{y}) = \text{Re}(\exp((\boldsymbol{W}_r + i\boldsymbol{W}_i)(\log \boldsymbol{r} + i\pi\boldsymbol{k}))) \tag{9}$$

$$= \exp(\boldsymbol{W}_r \log \boldsymbol{r} - \pi\boldsymbol{W}_i\boldsymbol{k}) \odot \cos(\boldsymbol{W}_i \log \boldsymbol{r} + \pi\boldsymbol{W}_r\boldsymbol{k}). \tag{10}$$

Above we have used Euler's formula $e^{ix} = \cos x + i \sin x$. A diagram of the NaiveNPU is shown in Fig. 1.

**Definition** (**NaiveNPU**). *The Naive Neural Power Unit, with matrices $\boldsymbol{W}_r$ and $\boldsymbol{W}_i$ representing real and imaginary part of the complex numbers, is defined as*

$$\boldsymbol{y} = \exp(\boldsymbol{W}_r \log \boldsymbol{r} - \pi\boldsymbol{W}_i\boldsymbol{k}) \odot \cos(\boldsymbol{W}_i \log \boldsymbol{r} + \pi\boldsymbol{W}_r\boldsymbol{k}), \textit{ where} \tag{11}$$

$$\boldsymbol{r} = |\boldsymbol{x}| + \epsilon, \quad k_i = \begin{cases} 0 & x_i \geq 0 \\ 1 & x_i < 0 \end{cases},$$

*with inputs $\boldsymbol{x}$, machine epsilon $\epsilon$, and learnt parameters $\boldsymbol{W}_r$ and $\boldsymbol{W}_i$.*

### 3.2 The Relevance Gate – NPU

The NaiveNPU has difficulties to converge on large scale tasks, and to reach sparse results in cases where the input to a given row is small. We demonstrate this on a toy example of learning the function $f : \mathbb{R}^2 \to \mathbb{R}$, which is the identity on one of two inputs. The task is defined by the loss $\mathcal{L}$:

$$\mathcal{L} = \sum_i |m(x_1, x_2) - f(x_1, x_2)| = \sum_i |m(x_1, x_2) - x_{1,i}|,$$

where $m = \text{NaiveNPU}$ with $(\boldsymbol{W}_r, \boldsymbol{W}_i) \in \mathbb{R}^{1 \times 2}$ \tag{12}

and $x_1 \sim \mathcal{U}(0, 2), x_2 \sim \mathcal{U}(0, 0.05)$.

The left plot in Fig. 3 depicts the gradient norm $\mathcal{G}$

$$\mathcal{G}(\boldsymbol{W}_r) = \left\| \frac{\partial \mathcal{L}}{\partial \boldsymbol{W}_r} \right\|_2 \tag{13}$$

of the NaiveNPU for a batch of two-dimensional inputs. Even in this simple example, the gradient of the NaiveNPU is close to zero in large parts of the parameter space. This can be explained as follows. One row of NaiveNPU weights effectively raises each input to a power and multiplies them: $x_1^{w_1} x_2^{w_2} \ldots x_n^{w_n}$. If a single input $x_i$ is constantly close to zero (i.e. irrelevant), the whole row will be zero, no matter what its weights are and the gradient information on all other weights is lost. Therefore, we introduce a gate on the input of our layer that can turn irrelevant inputs into 1s. A diagram of the NPU is shown in Fig. 2.

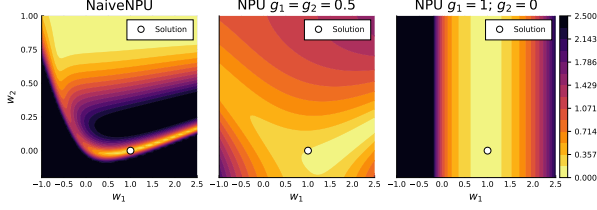

Figure 3: Gradient norm $\mathcal{G}$ of NaiveNPU and NPU for the task of learning the identity on $x_1$ (black areas are beyond the color scale). Inputs and loss are defined in Eq. 12. The correct solution is $w_1 = 1$ and $w_2 = 0$. The NaiveNPU has a large zero gradient region for $w_2 > 0.75$, while the NPU's surface is much more informative. The gates in the central plot are fixed at $g_1 = g_2 = 0.5$ which corresponds to the initial gate parameters. During training they adjust as needed, in this case to $g_1 = 1$ and $g_2 = 0$. $\boldsymbol{W}_i$ is set to zero in all plots.

**Definition (NPU).** *The NPU extends the NaiveNPU by the relevance gate $\boldsymbol{g}$ on the input $\boldsymbol{x}$.*

$$\boldsymbol{y} = \exp(\boldsymbol{W}_r \log \boldsymbol{r} - \pi \boldsymbol{W}_i \boldsymbol{k}) \odot \cos(\boldsymbol{W}_i \log \boldsymbol{r} + \pi \boldsymbol{W}_r \boldsymbol{k}), \; where \tag{14}$$

$$\boldsymbol{r} = \hat{\boldsymbol{g}} \odot (|\boldsymbol{x}| + \epsilon) + (1 - \hat{\boldsymbol{g}}), \quad k_i = \begin{cases} 0 & x_i \geq 0 \\ \hat{g}_i & x_i < 0 \end{cases}, \quad \hat{g}_i = \min(\max(g_i, 0), 1), \tag{15}$$

*with inputs $\boldsymbol{x}$, and learnt parameters $\boldsymbol{W}_r$, $\boldsymbol{W}_i$ and $\boldsymbol{g}$.*

The central plot of Fig. 3 shows the gradient norm $\mathcal{G}$ of the NPU on the identity task with its initial gate setting of $g_1 = g_2 = 0.5$. The large zero-gradient region of the NaiveNPU is gone. The last plot shows the same for $g_1 = 1$ and $g_2 = 0$, which corresponds to the correct gates at the end of NPU training. The gradient is independent of $w_2$, which means that it can easily be pruned by a simple regularization such as $L_1$. In Sec. 4.3 we show how important the relevance gating mechanism is for the convergence and sparsity of large models. Sparsity is especially important in order to use the NPU as a transparent model.

**Initialization** We recommend initializing the NPU with a Glorot Uniform distribution on the real weights $\boldsymbol{W}_r$. The imaginary weights $\boldsymbol{W}_i$ can be initialized to zeros, so they will only be used where necessary, and the gate $\boldsymbol{g}$ with 0.5, so the NPU can choose to output 1.

**Definition (RealNPU).** *In many practical tasks, such as multiplication or division, the final value of $\boldsymbol{W}_i$ should be equal to zero. We will denote NPU with removed parameters for the imaginary part as RealNPU and study the impact of this change on convergence in Sec. 4.*

## 4 Experiments

In Sec. 4.1, we show how the NPU can help to build better NODE models. Additionally, we use the RealNPU as a highly transparent model, from which we can directly recover the generating equation of an ODE containing fractional powers. Subsequent Secs. 4.2 and 4.3 compare the NPU to prior art (NALU and NMU) on arithmetic tasks typically used to benchmark arithmetic units.

### 4.1 A Step Towards Equation Discovery of an Epidemiological Model

Data-driven models such as *SINDy* [Champion et al., 2019] or *Neural Ordinary Differential Equations* (NODE, Chen et al. [2019]) are used more and more in scientific applications. Recently, *Universal Differential Equations* (UDEs, Rackauckas et al. [2020]) were introduced which aim to combine data-driven models with physically informed differential equations to maximize interpretability/explainability of the resulting models.

If an ODE model is composed of dense layers, its direct interpretation is problematic and has to be performed retrospectively. The class of models based on SINDy is transparent by design, however it can only provide explanation within a linear combination of predefined set of basis functions. Thus, it cannot learn models with unknown fractional powers. With this experiment we aim to show that

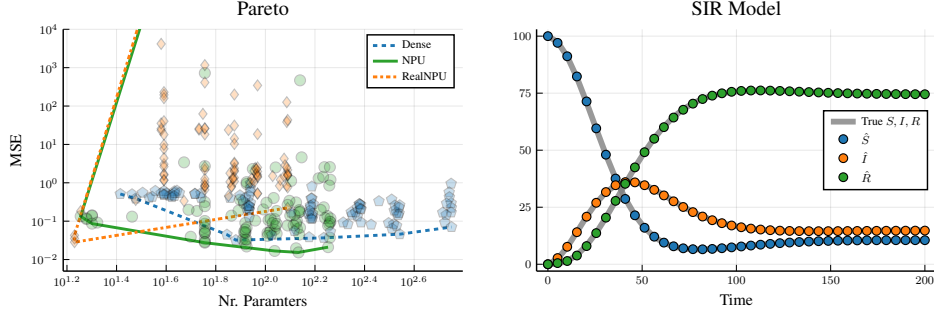

Figure 4: Pareto fronts of the dense network, NPU, and RealNPU. The NPU reaches solutions with lower MSE and fewer parameters than the dense net. The RealNPU mostly yields worse results than the NPU. Just in a few cases it converges to very sparse models with good MSE.

the NPU can *potentially* discover exact ODE models.[3] An example of an ODE that contains powers is a modification of the well-known epidemiological SIR model [Kermack et al., 1927] to fractional powers (fSIR, Taghvaei et al. [2020]), which was shown to be a beneficial modification for modelling the COVID-19 outbreak. The SIR model is built from three variables: $S$ (susceptible), $I$ (infectious), and $R$ (recovered/removed). Arguably the most important part of the model is the transmission rate $r$, which is typically taken to be proportional to the product of $S$ and $I$. Taghvaei et al. [2020] argue that, especially in the initial phase of an epidemic, the boundary areas of infected and susceptible cells scale with a fractional power, which leads to Eq. 17:

$$\frac{dS}{dt} = -r(t) + \eta R(t), \qquad \frac{dI}{dt} = r(t) - \alpha I(t), \qquad \frac{dR}{dt} = \alpha I(t) - \eta R(t), \qquad (16)$$

$$r(t) = \beta I(t)^\gamma S(t)^\kappa, \qquad (17)$$

We have numerically simulated one realization of the fSIR model with the parameters $\alpha = 0.05$, $\beta = 0.06$, $\eta = 0.01$, $\gamma = \kappa = 0.5$, in 40 time steps that are equally spaced in the time interval $T = (0, 200)$, such that the training data $\boldsymbol{X} = [S_t, I_t, R_t]_{t=1}^{40}$ contains one time series each for $S$, $I$, and $R$. The initial conditions $\boldsymbol{u}_0 = [S_0, I_0, R_0]$ are set to $S_0 = 100$, $I_0 = 0.01$, and $R_0 = 0$, as shown in Figure 4 (right). We fit the data with three different NODEs composed of different model types: a dense network, the NPU, and the RealNPU. An exemplary model is: $\mathrm{NPU} = \mathrm{Chain}(\mathrm{NPU}(3, h), \mathrm{NAU}(h, 3))$ with variable hidden size $h$. The detailed models are defined in Tab. A1. The training objective is the loss $\mathcal{L}$ with $L_1$ regularization.

$$\mathcal{L} = \mathrm{MSE}(\boldsymbol{X}, \mathrm{NODE}_\theta(\boldsymbol{u}_0)) + \beta ||\boldsymbol{\theta}||_1. \qquad (18)$$

We train each model for 3000 steps with the ADAM optimizer and a learning rate of 0.005, and subsequently with LBFGS until convergence (or for maximum 1000 steps). For each model type, we run a small grid search to build a Pareto front with $h \in \{6, 9, 12, 15, 20\}$ and $\beta \in \{0, 0.01, 0.1, 1\}$, where each hyper-parameter pair is run five times. The resulting Pareto front is shown on the left of Fig. 4. The NPU reaches much sparser and better solutions than the dense network. The RealNPU has problems to converge in the majority of cases, however, there are a few models in the bottom left that reach a very low MSE and have very few parameters. The best of these models is shown in Fig. 5. It looks strikingly similar to the fSIR model in matrix form:

$$\begin{bmatrix} \dot{S} \\ \dot{I} \\ \dot{R} \end{bmatrix} = \begin{bmatrix} -\beta & 0 & \eta \\ \beta & -\alpha & 0 \\ 0 & \alpha & \eta \end{bmatrix} \begin{bmatrix} I^\gamma S^\kappa \\ I \\ R \end{bmatrix}. \qquad (19)$$

Reading Fig. 5 from right to left, we can extract the ODE that the RealNPU represents. The first hidden variable correctly identified the transmission rate as a product of two fractional powers $r = I^\gamma S^\kappa$ with $\kappa = 0.57$ and $\gamma = 0.62$, which is close to the true values $\gamma = \kappa = 0.5$. The second, third, and the last hidden variable were found to be irrelevant (the relevance gate returns 1). The

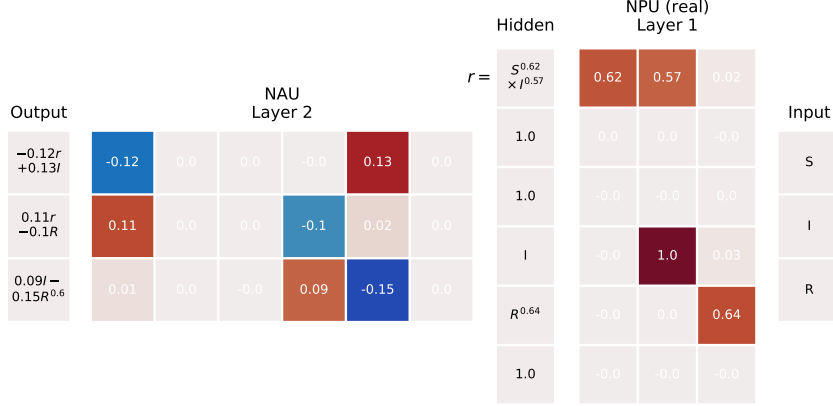

Figure 5: Visualization of the best RealNPU. Reading from right to left, it takes the SIR variables as an input, then applies the NPU and the NAU. It correctly identifies $r$ as a fractional product in the NPU, and gets the rest of the fSIR parameters almost right in the NAU.

fourth hidden variable is a selector of the second input $I$, and the fifth hidden variable is selector of a power of $R$, $R^{0.64}$. In the second layer, the NAU combines the correct hidden outputs from the NPU such that $\dot{S}$ is composed of the negative transmission rate $r$ and positive $R$. $\dot{I}$ and $\dot{R}$ are also composed of the correct hidden variables, with the parameters $\alpha, \beta, \eta$ being not far off from the truth. We conclude that even with this very naive approach, the RealNPU can recover something close to the true fractional SIR model.

In summary, the NPU can work well in sequential tasks, and we have shown that we can reach highly transparent results with the RealNPU, but in practice, using the RealNPU might be difficult due to its lower success rate. With a more elaborate analysis, it should be possible to reach the same solutions with the full NPU and e.g. a strong regularization of its imaginary parameters.

## 4.2   Simple Arithmetic Task

In this experiment we compare six different layers (NPU, RealNPU, NMU, NALU, iNALU, Dense) on a small problem with two inputs and four outputs. The objective is to learn the function $f : \mathbb{R}^2 \to \mathbb{R}^4$ with a standard MSE loss:

$$f(x,y) = (x + y,\ xy,\ x/y,\ \sqrt{x}\ )^T =: \boldsymbol{t}, \tag{20}$$

$$\mathcal{L} = \frac{1}{4}\sum_{i=1}^{4}\left(\text{model}(x,y)_i - f(x,y)_i\right) = \text{MSE}(\hat{\boldsymbol{t}}, \boldsymbol{t}). \tag{21}$$

Learning the function $f$ includes not only learning the correct arithmetic operation, but also to separate them cleanly, which tests the gating mechanisms of the layers. Each model has two layers with a hidden dimension $h$. E.g. the NPU model is defined by $\text{NPU} = \text{Chain}(\text{NPU}(2, h = 6), \text{NAU}(h = 6, 4))$. The remaining models that are used in the tables and plots are given in Tab. A3. To obtain valid results in case of division we train on positive, non-zero inputs, but test on negative, non-zero numbers (except for test inputs to the square-root):

$$(x_{\text{train}}, y_{\text{train}}) \sim \mathcal{U}(0.1, 2) \quad (x_{\text{test}}, y_{\text{test}}) \sim \mathcal{R}(\text{-}4.1{:}0.2{:}4) \quad (x_{\text{test, sqrt}}, y_{\text{test, sqrt}}) \sim \mathcal{R}(0.1{:}0.1{:}4) \tag{22}$$

where $\mathcal{R}$ denotes a *range* with start, step, and end. We train each model for $20\,000$ steps with the ADAM optimizer, a learning rate of 0.001, and a batch size of 100. The input samples are generated on the fly during training. Fig. 6 shows the error surface of the best of 20 models on each task. Tab. A2 lists the corresponding averaged testing errors of all 20 models.

Both NPUs successfully learn $(+, \times, \div, \sqrt{\cdot})$ and clearly outperform NALU and iNALU on all tasks. Surprisingly, the NALU has problems extrapolating in this task, which as Schlör et al. [2020] suggest, might be due to its gating mechanism. The NPUs are on par with the NMU for $(+)$, but the NMU is better at $(\times)$ due to its inductive bias. The NMU cannot learn $(\div, \sqrt{\cdot})$. The fact that the RealNPU performs slightly better than the NPU indicates that the task is easy enough to not require the

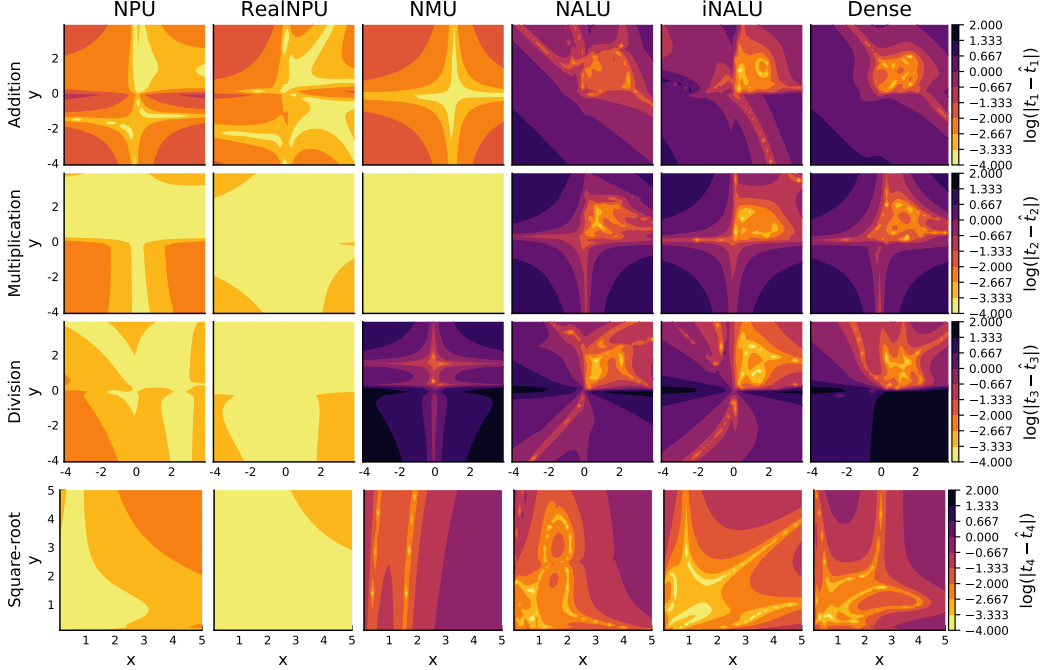

Figure 6: Comparison of extrapolation quality of different models learning Eq. 20. Each column represents the best model of 20 runs that were trained on the range $\mathcal{U}(0.1, 2)$. Lighter color implies lower error.

imaginary parameters to help convergence. In such a case, the RealNPU generalizes better because it corresponds to the task it is trying to learn.

## 4.3 Large Scale Arithmetic Task

One of the most important properties of a layer in a neural network is its ability to scale. With the large scale arithmetic task we show that the NPU works reliably on many-input tasks that are heavily over-parametrized. In this section we compare NALU, NMU, NPU, RealNPU, and the NaiveNPU on a task that is identical to the 'arithmetic task' that Madsen and Johansen [2020] and Trask et al. [2018] analyse as well. The goal is to sum two subsets of a 100 dimensional vector and apply an operation (like $\times$) to the two summed subsets. The dataset generation is defined in the set of Eq. 23, with the parameters from Tab. A5.

$$a = \sum_{i=s_{1,\text{start}}}^{s_{1,\text{end}}} x_i, \quad b = \sum_{i=s_{2,\text{start}}}^{s_{2,\text{end}}} x_i, \quad y_{\text{add}} = a + b, \quad y_{\text{mul}} = a \times b, \quad y_{\text{div}} = 1/a, \quad y_{\text{sqrt}} = \sqrt{a}, \quad (23)$$

where starting and ending values $s_{i,\text{start}}, s_{i,\text{end}}$ of the summations are chosen such that $a$ and $b$ come from subsets of the input vector $\boldsymbol{x}$ with a given overlap. The training objective is standard MSE, regularized with $L_1$:

$$\mathcal{L} = \text{MSE}(\text{model}(\boldsymbol{x}), y) + \beta \left\| \boldsymbol{\theta} \right\|_1, \quad (24)$$

where $\beta$ is scheduled to be low in the beginning of training and stronger towards the end. Specifics of the used models and their hyper-parameters are defined in Tab. A4 & A6. Madsen and Johansen [2020] perform an extensive analysis of this task with different subset and overlap ratios, varying model and input sizes, and much more, establishing that the combination of NAU/NMU outperforms the NALU. We focus on the comparison of NPU, RealNPU, NMU, and NALU on the default parameters of Madsen and Johansen [2020] which sets the subset ratio to 0.5 and the overlap ratio to 0.25 (details in Tab. A5). We include the NaiveNPU (without the relevance gate) to show how important the gating mechanism is for both sparsity and overall performance.

Fig. 7 plots testing errors over the number of non-zero parameters for all models and tasks. The addition plot shows that NMU, NPU, and RealNPU successfully learn and extrapolate on $(+)$ with

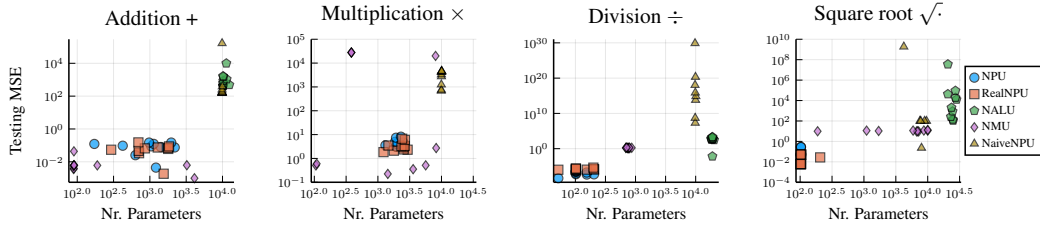

Figure 7: Testing MSE over number of non-zero parameters ($w_i > 0.001$) of the large scale arithmetic task. The NMU outperforms the NPU on its native tasks, addition and multiplication. The NPU is the best at division and square-root. The NaiveNPU without the relevance gate is far off, because it does not have the necessary gradient signal to converge, as discussed in Sec. 3.2

Table 1: Testing errors of the large scale arithmetic task. Each value is obtained by computing median (and median absolute deviation) of 10 runs.

| Task | NPU | RealNPU | NALU | NMU | NaiveNPU |
|---|---|---|---|---|---|
| $+$ | $0.092 \pm 0.031$ | $0.063 \pm 0.014$ | $740.0 \pm 330.0$ | $\mathbf{0.00602 \pm 0.00019}$ | $161.65 \pm 0.11$ |
| $\times$ | $4.28 \pm 0.9$ | $3.09 \pm 0.74$ | $2.9e83 \pm 2.9e83$ | $\mathbf{1.7 \pm 1.4}$ | $3750.0 \pm 870.0$ |
| $\div$ | $\mathbf{1.0e\text{-}7 \pm 1.0e\text{-}7}$ | $1.4e\text{-}6 \pm 4.0e\text{-}7$ | $530.0 \pm 200.0$ | $1.622 \pm 0.081$ | $5.4e17 \pm 5.4e17$ |
| $\sqrt{\cdot}$ | $0.054 \pm 0.0078$ | $\mathbf{0.017 \pm 0.011}$ | $7300.0 \pm 7200.0$ | $10.96 \pm 0.89$ | $9.3e8 \pm 9.3e8$ |

the NMU converging to the sparsest and most accurate models. On ($\times$), the best NMU models outperform the NPU and RealNPU, but some NMUs do not converge at all. The testing MSE of the NALU is so large that it is excluded from the plot. On ($\div$, $\sqrt{\cdot}$) the NPU clearly outperforms all other layers in MSE and sparsity. Generally, the difference between the NaiveNPU and the other NPUs is huge and demonstrates how important the relevance gate is both for convergence and sparsity. The NPUs with relevance gates effectively convert irrelevant inputs to 1s, while the NaiveNPU is stuck on the zero gradient plateau.

## 5 Conclusion

We introduced the *Neural Power Unit* which addresses the deficiencies of current arithmetic units: it can learn multiplication, division, and arbitrary power functions for positive, negative, and small numbers. We showed that the NPU outperforms its main competitor (NALU) and reaches performance that is on par with the multiplication specialist NMU (Sec. 4.2 and 4.3).
Additionally, we have demonstrated that the NPU converges consistently, even on sequential tasks. The RealNPU can be used as a transparent model that is capable of recovering the governing equations of dynamical systems purely from the data (Sec. 4.1).

## Broader Impact

Current neural network architectures are often perceived as black box models that are difficult to explain or interpret. This becomes highly problematic if ML models are involved in high stakes decisions in e.g. criminal justice, healthcare, or control systems. With the NPU, we hope to contribute to the broad topic of interpretable machine learning, with a focus on scientific applications. Additionally, learning to abstract (mathematical) ideas and extrapolating is a fundamental goal that might contribute to more reliable machine learning systems.

However, the inductive biases that are used to increase transparency in the NPU can cause the model to ignore subgroups in the data. This is not an issue for learning arithmetic operations, but could lead to biased models in more general use cases.

The methodology presented in the experiments in Sec. 4.1 is not indented to be used for real-world epidemiological predictions. They are merely demonstrating that the NPU can learn an ODE with

fractional powers. For an application of the NPU much more post-processing has to be done to ensure the reliability of the results.

## Acknowledgements and Disclosure of Funding

The research presented in this work has been supported by the Grant Agency of Czech Republic no. 18-21409S. The authors also acknowledge the support of the OP VVV MEYS funded project CZ.02.1.01/0.0/0.0/16_019/0000765 "Research Center for Informatics".

We thank the authors of the Julia packages Flux.jl [Innes et al., 2018] and DifferentialEquations.jl [Rackauckas and Nie, 2017]).

## Footnotes

[1]Implementation of Neural Arithmetic Units: github.com/nmheim/NeuralArithmetic.jl The code to reproduce our experiments is available at github.com/nmheim/NeuralPowerUnits.

[2]as defined by Lipton [2017]

[3]The demonstration given here is not intended to be used in practice. For real-world predictions in such a sensitive area much more post-processing is needed to ensure safe predictions.

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
