[Supplementary Material 1]

# Neural Power Units

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

Above we have used Euler's formula[1] and the fact that the complex logarithm for real valued inputs is

$$\log x = \log r + i\theta = \log r + ik\pi, \tag{10}$$

where $k = 0$ if $r \geq 0$ and $k = 1$ if $r < 0$. A diagram of the NaiveNPU is shown in Fig. 1.

**Definition (NaiveNPU).** *The Naive Neural Power Unit with matrices $\boldsymbol{W}_r$ and $\boldsymbol{W}_i$ representing real and imaginary part of the complex numbers defined as*

$$\boldsymbol{z} = \exp(\boldsymbol{W}_r \log \boldsymbol{r} - \pi \boldsymbol{W}_i \boldsymbol{k}) \odot \cos(\boldsymbol{W}_i \log \boldsymbol{r} + \pi \boldsymbol{W}_r \boldsymbol{k}), \text{ where} \tag{11}$$

$$\boldsymbol{r} = |\boldsymbol{x}|, \quad k_i = \begin{cases} 0 & x_i \leq 0 \\ 1 & x_i > 0 \end{cases}$$

## 3.2 The Relevance Gate – NPU

The NaiveNPU has difficulties to converge on large scale tasks, and to reach sparse results in cases where the input to a given row is small. We demonstrate this on a toy example of learning the identity on one of two inputs and neglecting the second one, $f(x_1, x_2) = x_1$:

$$\mathcal{L} = |\text{m}(x_1, x_2) - x_1|, \quad \text{where} \quad \text{m} = \text{NPU}(2, 1), \quad x_1 \sim \mathcal{U}(0, 2), \quad x_2 \sim \mathcal{U}(0, 0.05).$$

The left plot in Fig. 3 depicts the gradient norm of NPU and NaiveNPU for a batch of two-dimensional inputs. One input is small and irrelevant. Even in this simple example, gradient of the NaiveNPU is close to zero in large parts of the parameter space This can be explained as follows. One row of NaiveNPU weights effectively raises each input to a power and multiplies them: $x_1^{w_1} x_2^{w_2} \ldots x_n^{w_n}$. If a single input $x_i$ is constantly close to zero (i.e. irrelevant), the whole row will be zero, no matter what its weights are and the gradient information on all other weights is lost. Therefore, we introduce a gate on the input of the NPU that can turn irrelevant inputs into 1s. A diagram of the NPU is shown in Fig. 2.

**Definition (NPU).** *The NPU extends the NaiveNPU by the relevance gate $g$ on the input $x$.*

$$\boldsymbol{z} = \exp(\boldsymbol{W}_r \log \boldsymbol{r} - \pi \boldsymbol{W}_i \boldsymbol{k}) \odot \cos(\boldsymbol{W}_i \log \boldsymbol{r} + \pi \boldsymbol{W}_r \boldsymbol{k}), \text{ where} \tag{12}$$

$$\boldsymbol{r} = \hat{\boldsymbol{g}} \odot |\boldsymbol{x}| + (1 - \hat{\boldsymbol{g}}), \quad k_i = \begin{cases} 0 & x_i \leq 0 \\ \hat{g}_i & x_i > 0 \end{cases}, \quad \hat{g}_i = \min(\max(g_i, 0), 1) \tag{13}$$

The central plot of Fig. 3 shows the gradient of the NPU on the identity task with its initial gate setting of $g_1 = g_2 = 0.5$. The large zero-gradient region of the NaiveNPU is gone. The last plot shows the same loss for $g_1 = 1$ and $g_2 = 0$, which corresponds to the correct gates at the end of NPU training. The gradient is independent of $w_2$, which means that it can easily be pruned by a simple regularization such as L1. In Sec. 4.3 we show how important the relevance gating mechanism is for the convergence and sparsity of large models. Sparsity is especially important in order to use the NPU as an interpretable model.

**Initialization** We recommend initializing the NPU with a Glorot Uniform distribution on the real weights $\boldsymbol{W}_r$. The imaginary weights $\boldsymbol{W}_i$ can be initialized to zeros, so they will only be used where necessary, and the gate $g$ with 0.5, so the NPU can choose to output 1.

**Definition (RealNPU).** *In many practical tasks, such as multiplication or division, the final value of $\boldsymbol{W}_i$ should be equal to zero. We will denote NPU with removed parameters for the imaginary part as RealNPU and study the impact of this change on convergence in Sec. 4.*

Figure 3: Norm of the gradient of NaiveNPU and NPU for the task of learning the identity on $x_1$. Inputs and loss are defined on the right, gradient surfaces on the left (black areas are beyond the color scale).

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

[Supplementary Material 2]

**Appendix**

Table A1: Model definitions for the fSIR task.

| Model | Layer 1 | Layer 2 | Layer 3 |
|-------|---------|---------|---------|
| NPU | $\text{NPU}(3, h)$ | $\text{NAU}(h, 3)$ | – |
| NPU | $\text{NPU}_{\text{real}}(3, h)$ | $\text{NAU}(h, 3)$ | – |
| Dense | $\text{Dense}(2, h, \sigma)$ | $\text{Dense}(h, h, \sigma)$ | $\text{Dense}(h, 3)$ |

Table A2: Testing error on the simple arithmetic task for the different models (i.e. mean of each heatmap in Fig. 6). Each value is obtained by computing median (and median absolute deviation) of the error of 20 models.

| Task | NPU | RealNPU | NMU | NALU | iNALU | Dense |
|------|-----|---------|-----|------|-------|-------|
| $+$ | $0.2 \pm 0.11$ | $\mathbf{0.08 \pm 0.021}$ | $0.2 \pm 0.18$ | $2.69 \pm 0.22$ | $2.18 \pm 0.13$ | $2.103 \pm 0.04$ |
| $\times$ | $0.37 \pm 0.23$ | $0.066 \pm 0.026$ | $\mathbf{0.005 \pm 0.004}$ | $4.55 \pm 0.2$ | $3.453 \pm 0.065$ | $3.546 \pm 0.035$ |
| $\div$ | $0.23 \pm 0.13$ | $\mathbf{0.085 \pm 0.038}$ | $11.399 \pm 0.035$ | $3.33 \pm 0.18$ | $2.54 \pm 0.26$ | $14.16 \pm 0.23$ |
| $\sqrt{\cdot}$ | $0.031 \pm 0.025$ | $\mathbf{0.004 \pm 0.001}$ | $0.16 \pm 0.002$ | $0.034 \pm 0.006$ | $0.049 \pm 0.011$ | $0.084 \pm 0.007$ |

Table A3: Model definitions for the simple arithmetic task.

| Model | Layer 1 | Layer 2 | Layer 3 |
|-------|---------|---------|---------|
| NPU | $\text{NAU}(2, 6)$ | $\text{NPU}(6, 2)$ | – |
| RealNPU | $\text{NAU}(2, 6)$ | $\text{RealNPU}(6, 2)$ | – |
| NMU | $\text{NAU}(2, 6)$ | $\text{NMU}(6, 2)$ | – |
| NALU | $\text{NALU}(2, 6)$ | $\text{NALU}(6, 2)$ | – |
| iNALU | $\text{iNALU}(2, 6)$ | $\text{iNALU}(6, 2)$ | – |
| Dense | $\text{Dense}(2, 10, \sigma)$ | $\text{Dense}(10, 10, \sigma)$ | $\text{Dense}(10, 2)$ |

Table A4: Model definitions for the large scale arithmetic task.

| Model | Layer 1 | Layer 2 |
|---|---|---|
| NPU | NAU(100, 100) | NPU(100, 1) |
| NPU | NAU(100, 100) | NPU(100, 1) |
| NMU | NAU(100, 100) | NMU(100, 1) |
| NALU | NALU(100, 100) | NALU(100, 1) |

Table A5: Dataset parameters for the large scale arithmetic task.

| Task | Input size | Subset ratio | Overlap ratio | Training range | Validation range |
|---|---|---|---|---|---|
| Add | 100 | 0.5 | 0.25 | Sobol(-1,1) | Sobol(-4,4) |
| Mult | 100 | 0.5 | 0.25 | Sobol(-1,1) | Sobol(-4,4) |
| Div | 100 | 0.5 | – | Sobol(0,0.5) | Sobol(-0.5,0.5) |
| Sqrt | 100 | 0.5 | – | Sobol(0,2) | Sobol(0,4) |

Table A6: Training parameters for the large scale arithmetic task. The $\beta$-parameters define the stepwise exponential growth of the L1 regularization with start, step, growth, and end.

| Task | Learning rate | Iterations | $\beta_{start}$ | $\beta_{end}$ | $\beta_{step}$ | $\beta_{growth}$ |
|---|---|---|---|---|---|---|
| Add | 1e-2 | 1e5 | 1e-5 | 1e-4 | 10 000 | 10 |
| Mult | 5e-3 | 1e5 | 1e-5 | 1e-7 | 10 000 | 10 |
| Div | 5e-3 | 1e5 | 1e-9 | 1e-7 | 10 000 | 10 |
| Sqrt | 5e-3 | 1e5 | 1e-6 | 1e-4 | 10 000 | 10 |