[Reviews · NeurIPS 2020]

Review 1

Summary and Contributions: This paper considers NN architectures that can learn arithmetic operation meaningfully-- i.e. these are NNs that generalize to inputs beyond those used in the training phase. In other words, they really learn the general arithmetic operations. Previous work could either do the basic arithmetic operations (additions, subtraction, multiplication and division) but only for positive real numbers (this is the Neural Arithmetic Logic Unit work of Trask et al.) or they can work on entire range of real numbers but cannot handle division (e.g. work of Madsen and Johansen). The paper presents an NN architecture that can learn all four basic arithmetic operations as well as the powering operation for the entire range of real numbers. The paper exhibits this via three experimental results: (i) Equation discovery of an epidemiological model. In particular, the paper learns the classical SIR model (or rather a modification called fractional SIR model that has been used in COVID-19 modeling) where the transmission rate depends on three variables that evolve via differential equations. The paper picks a certain setting of parameters of the fSIR model and trains on time series data. The paper shows that the best learned model in some sense "recovers" the correct equation. (ii) Simple arithmetic task of computing a function that maps two numbers to the 4-tuple of their sum, product, ratio and square root (of the first number). The training set are random positive pairs. Testing of the first three operation are with negative numbers as well. For the square root problem the testing is on a larger range of numbers than those it was trained on. Error is measured by MSE. On all but the multiplication task the architecture proposed in this paper improves upon (for division and square root) existing work or is competitive on addition with existing work. (iii) Large scale arithmetic task used in previous work, where one has to first add ranges of numbers from two lists and then add/multiply/take reciprocal and square root of the first sum. Again the architectures proposed in this paper outperform existing architectures for division and square root. However, existing architectures are more competitive for addition and multiplication. The key technical insight in this work is the following (and we only focus on multiplication). In existing work one takes the log of the absolute values of numbers, adds them up and then exponentiates them. The problem is that this does not work well for values close to 0 (and existing work has to add a small positive \eps before taking log). In this work, the authors do a similar trick but first lift the numbers over complex numbers and then use logarithm over complex numbers. This allows them to not have to take the absolute value of the numbers (which allows the scheme in this paper to work over all real numbers). Actually since the work is for real numbers then take the real part of computing over complex numbers. The authors also propose a "gated" version of this architecture to help scale better for large sparse models.

Strengths: (*) The problem considered in the paper is natural and very well-worth studying and should be of interest to the wider NeurIPS community. (*) The paper tackles the division and powering operation, which was something that was lacking in previous work. (*) The paper uses the large scale arithmetic task from previous work and shows better performance of the proposed architecture for division and square root.

Weaknesses: All three experiments make some setup decisions that are not clearly justified. We tackle them in the order of their appearance in the paper: (i) The equation discovery of fSIR is motivated from its COVID use and the claim is that the architecture proposed in this paper "learns" such a model. However, there are several shortcomings that are not explained. (1) First, the paper picked specific values of \alpha, \beta, \eta, \gamma and \kappa. However, *no* justification is given for this choice-- are these parameters of interest to a specific instantiation of the model? Even if this choice is motivated by practice, I find it hard to believe that this means that the system proposed in this paper actually learns the model correctly. Since there are *no* results for any other settings of the parameters, it is not clear why the method proposed in the paper actually learning something general because there are no experiments for other settings of the parameters. (2) Second, the recovery is not exact. In the experiment the paper uses \gamma=\kappa = 0.5 but recovers the value \kappa=0.57 and \gamma=0.62. But there is *no* justification for why this is "close enough". I.e. there is no baseline to determine what is a reasonable amount of error. As a sort of misleading analogy, if the actual R0 of COVID-19 is 1.05 and a model estimates it to be 0.95 then the outcomes are vastly different. Of course this is where having the wrong base can have very different consequences and the error here is in the exponent value but my point is that there is no way to judge how serious is the 24% error in estimating the value of \gamma. (3) Finally, I found Figure 5 and the corresponding claim about recovering the correct structure of the diff eq misleading since in layer 1, the paper ignores some exponents of 0.02 and 0.03 (which if included would make the final dependence of transmission on SIR different). Perhaps the idea is to zero out small values but again there is no way of knowing whether replacing 0.03 by 0 is a big change or not. In summary, I am not convinced about anything from this experiment. (There are also claims about interpretability, which I think might be overblown as well-- see the addition feedback section for more. However, this piece did not affect my evaluation of the paper.) (ii) The results for the simple arithmetic task are more convincing than the first one but even here I have some concerns. In particular, the training and testing are essentially done on *one* specific range. What happens if the ranges were chosen differently? How does the MSE change? Again I find it hard to see the progression from showing the result on one setting and then claiming that the NN architectures actually *learns* the arithmetic operations. Perhaps this setup is standard but there is no indication that this is the case. However, to be fair to the authors, this set of experiments is much more convincing than the first one. (iii) The last of experiments is most convincing: because it uses an existing benchmark to assess the efficacy of the proposed NN architecture. However, even in this experiment the paper makes a change from the setup in Madsen and Johansen that is not explained. After one computes the sum of appropriate ranges in Madsen and Johansen one of the goals is to compute the ratio of the two sums while this paper only computes the reciprocal of the first sum. There is no justification for why this change was made and/or why this change does not matter. In summary, there are concerns about all the three experiments in the paper.

Correctness: See the comments on weakness on the experimental setup above.

Clarity: The paper could definitely have been better written. See the additional feedback section for some specific comments but here are two general things: no where in the paper is there a clear statement of what problem the NPU is trying to solve. One can try and guess but a paper should clearly state the problem setup. Further, the figures are too small to see in original size. I'm guessing this was done partly to get within the page limit but some effects could be misleading-- e.g. the "zeroing" out of small values in layer 1 in Fig 5 is just not possible to catch unless one is looking at the magnified versions of Figure 5.

Relation to Prior Work: Prior work has been appropriately compared with.

Reproducibility: Yes

Additional Feedback: I'll begin with a general comment on the claim on interpretability as well as on the broader impact section. Then I'll list more minor comments (that hopefully help improve the presentation of the paper). (*) The paper seems to equate sparsity with interpretability. While this might have been one of the initial assumptions in the literature, notions of interpretability is more complicated. See the following paper by Lipton for more on this-- https://arxiv.org/abs/1606.03490 (*) I have two comments on the broader impact section (outside of my comment on interpretability above). First, while being able to abstract out in ML systems might improve reliability in certain metrics, it can in many cases lead to biased systems-- the high level idea of this claim being is the following. If one abstracts out the data then one is by design throwing away information specific to sub-groups, which can lead to systems biased against such sub-groups. To be fair the paper is talking about arithmetic operation so this criticism might not apply to this paper. The second comment is that the paper does not state anything about negative impacts and in fact I can envision one potential negative impact. This is related to the experiment related to COVID-19. As mentioned earlier there is no indication of how significant the error in determining the parameters of the system is. Here is a potential downside of the paper written as is: if the error is indeed significant and some goes ahead and trains an fSIR model using the architecture presented in this paper and then uses it in real world, it could end up with potentially bad predictions. Perhaps it not the intention of the authors that their system be used to learn an fSIR model in the real world but if that is the case then this should be *clearly* stated in the broader impact section. In general, once should think of the Broader Impact section as a call to think about how someone might *un-intentionally* mis-use your systems-- warnings about how your system could be used/implemented incorrectly are then a useful service to the broader community. Below are some detailed comments: [Eq (7)] The r in W_r should not be bold [Before equation (8)] Would be good to clearly state that x= r*e^{i\theta} [Display equation below line 130] "NPU" has not been defined yet. Further, no where in the paper does it define what the two arguments to NPU represent. [Eq (16)] Might be useful to explain the NODE_\theta(u_0) notation (in particular, how does it relate back to the SIR learning etc.) [Fig 4, left panel] Having the powers of 10 go up in increments of 0.2 is somewhat non-standard. [Table 1] it would be good to state the number of parameters required by various architectures for the reported testing error rate. ====== Post rebuttal comment======= I thank the authors for the extra explanations on why the experiments were setup the way they were in their rebuttal. However, the "downplaying" of the experimental results makes the overall impact of the paper weaker. I interpreted (with kind help of my co-reviewers) the authors' explanation that learning division in all ranges etc. is hard so this paper shows that at least in _some_ settings the techniques in the paper can do better with division as compared to existing work. So in that sense the paper clearly shows advancement of the state-of-the-art. However, what is missing is showing that this advancement leads to improvement somewhere else (I suppose the COVID-19 result was trying to do that but as the authors agree that that experiment has its own potential issues). In other words, I am not fully convinced that based on the experiments that are provided in the paper that the improvement provided for division is strong enough to actually be useful in other applications (at least not in the current incarnation of the results). In summary, my rating of the paper has moved from 4 to a 5 based on the authors' clarifications on their experimental setup but I still have reservations (as mentioned above) which means I could not rate this a 6 or higher.


Review 2

Summary and Contributions: This manuscript extends Neural Arithmetic Units, by providing a neural framework that is capable of learning arbitrary power functions in a single layer (including negative powers, enabling division operations).

Strengths: This manuscript introduces a novel arithmetic layer, which converges more stably than previous arithmetic units, and provides an interesting evaluation where it is shown that the approach can be used for equation-discovery in dynamical systems, with fractional powers.

Weaknesses: General: There is a concern about novelty, since the main novelty described in the manuscript results from changing the real log to a complex log. Moreover, for only nominal improvements over prior works -- on only half of the tasks -- the NPU still required stacking with the pre-existing NAU layer, in order to capture the full spectrum of simple arithmetic operations. Section(s): Introduction: It would be great if the manuscript would also provide examples of how NAUs/NPUs may indeed be combined with “standard” neural networks, for a series of common tasks, on which performance of those “standard” neural networks remains poor. Section(s): The experimentation in the manuscript would be made stronger if equation-discovery were shown for a "complex engineered system" (e.g., differential flow in an automotive fuel-injection system, non-linear dynamics in commercial building HVAC control, smart-grid transactional energy, etc.), for which the first-order mathematical principles are readily verifiable. ### After reading author feedback and fellow reviewer comments ### I thank the authors for their feedback and addition insight on their experimental design. However, I am unable to increase my score, due to lasting concerns over the impact, applicability, and novelty of the work.

Correctness: The empirical methodology would be strengthened if more experimentation was provided: e.g., where NPU layers are used in otherwise "standard" neural networks, for typical tasks that require arithmetic operations (such as counting): natural language understanding, visual commonsense reasoning, program synthesis, etc.

Clarity: The paper was well-written

Relation to Prior Work: Yes

Reproducibility: Yes

Additional Feedback:


Review 3

Summary and Contributions: This paper presents the Neural Power Unit, a new arithmetic layer that can learn products of power functions, including multiplication and division. (Division, in particular, is not handled by previous layers.) This layer has improved convergence compared to NAUs in particular. The authors also present a real (as opposed to complex) version of this layer, which is more interpretable but as a tradeoff has more difficulty converging than the “base” NPU.

Strengths: This work presents an interesting contribution to the literature, following the recent trend of including structured layers (e.g. arithmetic or optimization layers) in neural networks in order to present useful inductive bias to the problem at hand. In the space of difference arithmetic units in particular, previous units have suffered convergence issues and have not been able to handle division. The proposed NPU seems to address the convergence issue and handle division. The presentation of the RealNPU is also interesting, and the interpretability vs. convergence tradeoff is well-explained. The empirical evaluation seems reasonable, with a variety of tasks at different complexities tested. The NPU performs well, and is only surpassed in one task that is shown (NMU, on the multiplication task — which makes sense given the specialized nature of the NMU).

Weaknesses: While I am unsure as to whether these present limitations, I have several questions regarding the experimental evaluation: * In Figure 4, I am not sure why the Pareto front has a huge spike in MSE for both the NPU and the RealNPU when the number of parameters is small. Could the authors shed some light on this? * In the results presented in Figure 6, why is NALU failing on the addition task, given that it is specialized for addition? * Why was training done with ADAM and then LBFGS (lines 184--185), and what other approaches were tried? How does this choice of training procedure affect the performance of each model? * In the simple arithmetic task, why was the choice made to train on positive (non-zero) inputs but test on negative (non-zero) inputs (as described in lines 212--213)? * How were the test architectures for the experimental baselines (given in the appendix) determined? Regarding the definition of the NPU: It seems that k can be something other than 0 or 1 (given that g_i can be fractional), which was not true for the NaiveNPU. In other words, it seems that real inputs to the NPU layer may be treated as complex due to the introduction of relevance gating. What are the implications of this, for both the base and real versions of the NPU? While the experiments shown are on synthetic arithmetic tasks, it could be useful to demonstrate the use of this layer on some “real-world” task in which arithmetic inductive bias is useful.

Correctness: The method and claims seem correct.

Clarity: The paper is well-written. In general, however, I think clarity could be improved a bit if relevant variable definitions were moved before the equations where they are used, rather than after. (E.g., in Equations 8 and 9, r and \theta are not defined until after those equations, and even then, they are only implicitly defined.)

Relation to Prior Work: Yes, the paper is well-contextualized with respect to previous work.

Reproducibility: Yes

Additional Feedback: Broader impacts: It is important that the authors evaluate any potential negative aspects of their work, alongside the positive aspects. Minor typos (not affecting my score): * Line 54: outperform —> outperforms * Line 125--126 (NaiveNPU definition): “numbers defined as” —> “numbers is defined as” * Line 133: missing a period * Caption in Figure 4: comma splice in second sentence Minor comments (not affecting my score): * Figure 5 might be easier to read if going from left to right (in a direction consistent with the reading of the text)


Review 4

Summary and Contributions: The authors propose a new arithmetic unit called the "neural power unit" (NPU) that aims at overcoming deficiencies in current arithmetic units to learn basic arithmetic operations that can generalize beyond the data seen during training and which allow for learning addition, multiplication, division and computation of square roots of real numbers. In particular, the paper makes three major contributions: 1) It introduces a novel arithmetic layer called the neural power unit. 2) It introduces a "relevance gating" mechanism that smoothes out the loss surface and significantly improves convergence. 3) The power unit is shown to be interpretable as demonstrated by its ability to identify an SIR compartmental model used in epidemiology to model the dynamics of an epidemic.

Strengths: The paper is very well written and provides a detailed and compelling analysis of the NPU. The shortcomings of the first version of the neural power unit (the NaiveNPU) are addressed and overcome by introducing the relevance gating mechanism. The level of detail provided in this paper seems sufficient to allow for reimplementing NaiveNPU, RealNPU and NPU and also take advantage of the newly introduced relevance gate.

Weaknesses: RealNPU may have a lower success rate than NPU as outlined as briefly addressed in lines 202-205. Overall however, this does not seem to be a major issue, and further refinements of the training procedure might overcome this.

Correctness: All claims made in this paper look correct to me. The math is sound and the results are compelling and advancing the state-of-the-art on the investigated tasks.

Clarity: This is an exceptionally well written paper that strikes the right balance of guiding the reader through the new material, placing it into the context of prior work and explaining the contributions and insights, thus being of great value to beginners and experts in the task of learning algebra and ordinary differential equations using neural networks.

Relation to Prior Work: Section 2 covers a comprehensive list of related and relevant studies. More importantly, this section reviews the advantages and weaknesses of the strongest models prior to this submission that the new model aims to overcome. The detailed review of NALU NMU provides a very clear picture of what the state-of-the-art had been.

Reproducibility: Yes

Additional Feedback: I have only one minor editorial suggestion: The sentence in line 205 is incomplete, ending in "because". I am using this box to amend my review. As stated before, I like this paper a lot and stay with my original assessment. I think that this paper has made tremendous progress in a space that is notoriously hard for NN to handle. There are many papers that are laborious to read, while very few are a true pleasure to review. This one belongs to the latter.

[Author Response · NeurIPS 2020]

First and foremost, we want to thank all the reviewers for their thorough reviews. We are especially happy about the high scores we have received from *Reviewer #3* and *Reviewer #4*, but also appreciate the in-depth analysis by *Reviewer #1* and the suggestions of *Reviewer #2*.

The problem we (and the NPU) are solving is to learn a subset of arithmetic operations (AOs), namely $(\times, \div, x^w)$. Prior art either has problems with small and negative numbers (NALU) or implements only a subset of AOs (NMU). We have demonstrated these limitations in experiments 2 & 3. *Reviewer #2* was concerned about sufficient novelty because we (just) perform the algebraic operations in *complex* rather than *real* space. We believe that it is precisely this simplicity, which makes our approach convincing, because it does not require additional, more complicated mechanisms to work (like e.g. iNALU), and it fixes the shortcomings of prior work e.g. by a mathematically correct treatment of negative inputs. *Reviewer #3* is correct that relevance gates that are not exactly zero or one lead to a complex treatment of real inputs, which introduces an error in the result. We believe this error is actually an advantage because the optimization has to push the gate to either zero or one to remove the error. However, we agree that gates should converge to exactly zero or one, and we plan to address this in future work (e.g. by an additional penalization).
In the following three paragraphs, we address the criticism expressed concerning our experiments as they appear in the paper.

**Experiment 1 - Equation Discovery**    Since the NALU reported problems with convergence, this experiment was set to demonstrate that the NPU can learn models of sequential data, which is notoriously difficult due to vanishing/exploding gradients, and bifurcations (also faced e.g. by RNNs). Additionally, small changes in the parameters can lead to substantial changes in the output after a few time steps. The experiment shows that the NPU can be used for sequential tasks, which we believe is owed to the combination of complex weights and the relevance gate.
Our second intention with this experiment was to outline how to utilize the transparency of the NPU as one part within a broader equation discovery framework. Since the NPU can represent more AOs than prior art, it is possible to fit the same data with simpler models (in terms of the number of parameters). This improves *transparency by decomposability* (see Lipton), which enables to extract a *practically useful* equation containing only a few terms. In the paper, we called this *interpretability*, which we will correct as suggested by *Reviewer #1*. We did not claim to recover the correct fSIR model, but rather something close to the original model. Finalization of the equation discovery would require more post-processing such as a detailed search of models near the obtained result, e.g., by using methods from binary neural networks or the regularization by penalizing entropy, as used in the GNN explainer. Considering the feedback on this experiment, we should have been more explicit about its purpose. We propose to change the title to "*A Step Towards Equation Discovery*", and to clarify in the text that the experiment is a proof of concept.
In response to *Reviewer #3*: The large MSE that some NPU realizations exhibit in Fig. 4 is caused by diverging models at the beginning of training (due to the difficulties named above), from which the models sometimes do not recover. We used ADAM to obtain a good initial solution further refined by LBFGS, which can improve MSE by around an order of magnitude.

**Experiment 2 - Simple Arithmetic**    In this experiment, we wanted to show that the NPU can learn fractional power functions, and additionally, that it finds a solution where the NALU fails. While the NALU can learn $+$, $\times$, and $\sqrt{\cdot}$ in isolation (see Trask et al.), it does not converge for the more complicated task of learning those AOs in one model, as in our setup of this task. Schlör et al. suggested that this might be due to the gating between addition and multiplication paths in the NALU. We only tested one range in this experiment, because $\div$ and $\sqrt{\cdot}$ quickly become approximately linear away from zero, making the gradient signal to the correct solution (i.e., equation) weak. Training models on such a weak gradient signal is very sensitive to step-size in SGD and requires increasing the batch sizes. We have therefore considered training on a range with stronger gradient signals to be a fairer comparison between models.

**Experiment 3 - Large Scale Arithmetic**    In this experiment, we aimed to stay close to the established benchmark by Madsen & Johansen. We have slightly modified it since the original setup of the task makes it difficult to learn $\div$, which is again due to weak gradient signals at the effectively sampled ranges. Specifically, their experiment applies a single AO to the sum of two overlapping subsets of a vector. For example, summing 50 numbers from a vector with samples from $\mathcal{U}(0, 1)$ - via the central limit theorem - results in samples for the AO of interest from a narrow Gaussian centered at $\mu = 25$. For $+$ and $\times$, which were the interest of the NMU experiments, the signal is still strong, but for $\div$, this is results in a weak signal and thus a very hard task for optimization. Therefore, we chose an easier inverse $x^{-1}$, which also proves our point (that the NPU can represent and learn division by a variable).

Finally, we would like to thank the reviewers again for their effort and offer to update our description of experiment 3 as stated above, and to add a negative example to the broader impact section (the misuse of the setup in experiment 3 for real-world COVID-19 predictions) in case our paper should be accepted.

[Meta-Review · NeurIPS 2020]

The reviewers agree that the paper made a significant contribution using the relevance gating mechanism mechanism to enhance the design of neural power unit. Empirical studies further support the power of NPU. Moreover, the trade off between interpretability and convergence is well-explained.